nature
microbiology
# Adaptation of the small intestine to microbial enteropathogens in Zambian children with stunting

Beatrice Amadi[1], Kanekwa Zyambo [1], Kanta Chandwe [1], Ellen Besa [1], Chola Mulenga[1], Simutanyi Mwakamui [1], Stepfanie Siyumbwa [1], Sophie Croft[2], Rose Banda[1], Miyoba Chipunza[1], Kapula Chifunda[1], Lydia Kazhila[1], Kelley VanBuskirk[1] and Paul Kelly [1,2] ✉

Environmental enteropathy is a major contributor to growth faltering in millions of children in Africa and South Asia. We carried out a longitudinal, observational and interventional study in Lusaka, Zambia, of 297 children with stunting (aged 2–17 months at recruitment) and 46 control children who had good growth (aged 1–5 months at recruitment). Control children contributed data only at baseline. Children were provided with nutritional supplementation of daily cornmeal-soy blend, an egg and a micronutrient sprinkle, and were followed up to 24 months of age. Children whose growth did not improve over 4–6 months of nutritional supplementation were classified as having non-responsive stunting. We monitored microbial translocation from the gut lumen to the bloodstream in the cohort with non-responsive stunting ($n = 108$) by measuring circulating lipopolysaccharide (LPS), LPS-binding protein and soluble CD14 at baseline and when non-response was declared. We found that microbial translocation decreased with increasing age, such that LPS declined in 81 (75%) of 108 children with non-responsive stunting, despite sustained pathogen pressure and ongoing intestinal epithelial damage. We used confocal laser endomicroscopy and found that mucosal leakiness also declined with age. However, expression of brush border enzyme, nutrient transporter and mucosal barrier genes in intestinal biopsies did not change with age or correlate with biomarkers of microbial translocation. We propose that environmental enteropathy arises through adaptation to pathogen-mediated epithelial damage. Although environmental enteropathy reduces microbial translocation, it does so at the cost of impaired growth. The reduced epithelial surface area imposed by villus blunting may explain these findings.

Mortality in African children under five years of age has fallen over recent decades[1,2] and much residual mortality is related to neonatal disorders or undernutrition[3]. Undernutrition, however, presents a frustrating paradox. It does not respond reliably to the provision of extra nutrients. This is most clearly seen in children with stunting, in whom nutritional supplementation consistently corrects only about 10% of the linear growth deficit[4–7]. Nutritional supplementation before or during pregnancy also delivers only a modest improvement in child linear growth[8]. Evidence now suggests that the refractory nature of linear growth failure is largely attributable to enteropathy present in undernutrition disorders[9–12]; enteropathy would also explain historical data regarding stunting[13].

The term enteropathy refers to a global change in mucosal structure and function in the small intestine. While there is no consensus on the drivers of environmental enteropathy, some contributors to its pathogenesis are understood. At an ecological level, it is related to socio-economic conditions rather than latitude[14], which is the principal reason for the renaming from 'tropical' to 'environmental'. Onset begins after birth[9,15]. It is seasonal[16], which strongly implicates environmental exposure, but the precise environmental noxae are uncertain. Environmental enteropathy resolved in Peace Corps volunteers after repatriation to the USA[17] and resolved in immigrants to the UK in proportion to time since arrival[18].

Enteropathogens (bacterial, protozoal and viral) have highly plausible roles as contributors to poor growth[19,20]. They cause epithelial damage, which is observed in children with growth failure[21]. Epithelial damage is a crucial step in enabling microbial translocation, which causes mucosal and systemic inflammation[20].

In this study, we report that microbial translocation gradually diminishes in intensity despite ongoing stunting, which we interpret as adaptation to pathogen-mediated epithelial damage.

## Results

**Description of the BEECH study.** The Biomarkers of Environmental Enteropathy in Children (BEECH) study was a community-based longitudinal study of stunting in the Misisi, Chawama, Kuku and John Laing residential areas in Lusaka, which took place from August 2016 to June 2019. Children ($n = 5,660$) between 0 and 18 months of age were screened for low weight-for-age; 401 were identified as eligible because length-for-age (LAZ), weight-for-age (WAZ) or weight-for-length (WLZ) scores were <−2; 297 were recruited when their primary caregiver gave written consent. Children were then followed up every 2 weeks; 213 were still under follow-up at 24 months of age (Extended Data Fig. 1). All children were entered into a nutritional rehabilitation programme that provided a daily ration of high-energy protein supplement (corn-soy blend) porridge, micronutrient sprinkles[22] (Nutromix; Hexagon Nutrition) and an egg[23]. The trajectories of LAZ and WLZ scores were evaluated after initiation of nutritional supplements. Those children ($n = 191$) whose LAZ scores were consistently <−2 over 4–6 months of observation were declared non-responders and underwent medical evaluation. In most children, no clinical explanation for non-responsive stunting (for example, cardiac disorder or tuberculosis) was found; endoscopy was offered in an attempt to find treatable causes for stunting. A total of 119 caregivers consented to endoscopy (Extended Data Fig. 1). At baseline, 69 children had wasting (WLZ, <−2) but by the date when non-response

[1]Tropical Gastroenterology and Nutrition Group, University of Zambia School of Medicine, Lusaka, Zambia. [2]Blizard Institute, Barts and The London School of Medicine and Dentistry, Queen Mary University of London, London, UK. ✉e-mail: m.p.kelly@qmul.ac.uk

**Table 1 | Characteristics of groups at baseline or when non-response was declared**

| Characteristics | Normal or expected range for biomarkers | Cases at baseline (n = 297) | Controls at baseline (n = 46) | Non-response declared (n = 118) | Pᵃ | Pᵇ |
|---|---|---|---|---|---|---|
| Sex (male:female) | | 156:141 | 25:21 | 59:59 | 0.66 | – |
| Age, months, median (IQR and range) | | 11 (7-14, 2-17) | 3 (3-5, 1-5) | 18 (15-21) | <0.0001 | |
| Anthropometry: LAZ[c] | | −2.6 | −0.9 | −3.3 | 0.0001 | <0.0001 |
| | | (−3.1, −2.2) | (−1.3, −0.4) | (−3.9, −2.8) | | |
| Anthropometry: WLZ[c] | | −1.3 | 0.8 | −0.7 | 0.0001 | <0.0001 |
| | | (−1.9, −0.7) | (0.2, 1.3) | (−1.3, −0.2) | | |
| Anthropometry: WAZ[c] | | −2.5 | −0.7 | −2.3 | 0.0001 | <0.0001 |
| | | (−3.0, −2.2) | (−0.5, 0.4) | (−2.7, −1.8) | | |
| Breastfeeding (n, %) | | 188 out of 291 (65%) | 19 out of 45 (42%) | 53 out of 116 (46%) | 0.005 | – |
| Diarrhoea incidence (measured prospectively by two-weekly recall) | | 5.7 episodes per year | 5.4 episodes per year | NA | 0.71 | – |
| Antibiotic use in previous two weeks at recruitment | | 84 (28%) | 6 (20%) | NA | 0.28 | – |
| HIV status | | Of 291: | Of 45: | Of 117: | 0.009 (trend test) | – |
| Unexposed | | 198 (68%) | 39 (87%) | 77 (66%) | | |
| Exposed but uninfected | | 83 (29%) | 6 (13%) | 38 (32%) | | |
| Infected | | 10 (3%) | 0 | 2 (2%) | | |
| LPS (EU ml⁻¹) | Not established | 402 (259-609) | 156 (121-252) | 191 (0-327) | <0.001 | <0.001 |
| LBP (ng ml⁻¹) | Not established | 15.8 (10.3-25.4) | 11.2 (8.6-14.9) | 8.2 (5.7-13.6) | <0.001 | 0.006 |
| sCD14 (mg l⁻¹) | 0.8-3.2 | 3.3 (1.9-4.6) | 1.6 (1.4-2.0) | 1.4 (1.2-2.2) | <0.001 | <0.001 |
| iFABP (ng ml⁻¹) | ULN = 0.224 (ref. [48]) or 0.450 (ref. [49]) | 0.84 (0.35-1.64) | 0.68 (0.25-1.20) | 1.90 (0.96-3.04) | 0.25 | 0.01 |

ᵃP values refer to children with stunting versus controls at baseline (two-sided Kruskal–Wallis test). ᵇP values refer to children with stunting at baseline and then at non-response (Wilcoxon signed-rank test). ᶜMedian and IQR. ULN, upper limit of normal.

was declared, only 9 had wasting; thus, wasting responded better to nutritional intervention than stunting. Controls (n = 46) were simultaneously recruited by identifying children with good growth from the same community (Table 1).

**Epithelial damage, microbial translocation and inflammation.** Circulating concentrations of bacterial lipopolysaccharide (LPS), LPS-binding protein (LBP) and soluble CD14 (sCD14) were greater in children with stunting at baseline than in controls (Fig. 1; P = 0.0001 by Kruskal–Wallis test for all). Intestinal fatty acid-binding protein (iFABP) concentrations did not differ (Fig. 1). Since controls were younger than children with stunting at baseline, we confirmed that differences in LPS, LBP and sCD14 concentrations were statistically significant even if restricted to children under 9 months of age (P < 0.001, P = 0.007 and P < 0.001, respectively by Kruskal–Wallis test). However, circulating biomarker concentrations in children with non-responsive stunting thereafter showed two markedly divergent patterns: iFABP concentrations increased over the period from baseline to the age when non-response was declared; circulating LPS, LBP and sCD14 concentrations decreased in the same children with non-response (P < 0.001, P = 0.006 and P < 0.001, respectively by Wilcoxon signed-rank test; Fig. 1). Decreases in LPS, LBP and sCD14 were observed in 75, 64 and 86% of children, respectively, while iFABP increased in 63% (Extended Data Fig. 2). Since obviously age increased between baseline and non-response, we next analysed changes with age in biomarker concentrations. Biomarkers of microbial translocation (LPS, LBP, sCD14) were reduced when non-response was declared, irrespective

of age, but iFABP concentrations were not reduced (Fig. 2). LAZ did not vary with age in children with non-responsive stunting but LAZ was inversely associated with iFABP concentrations (Extended Data Fig. 3). Biomarkers were not significantly different in the nine children with residual wasting on top of non-responsive stunting (data not shown).

**Enteropathogen colonization.** We detected enteropathogens considerably more frequently in children with stunting at baseline than in controls (Supplementary Table 1); children with stunting excreted up to 11 pathogens concurrently. Children with stunting under 9 months of age had a median of 4 (interquartile range (IQR) = 3–6) pathogens simultaneously; controls in this age band had only 1 (IQR = 0–3) pathogen (P = 0.0001). Pathogen detection did not diminish with age (Fig. 3a). Linear regression showed that age increased pathogen burden and confirmed that controls had two fewer pathogens per child (Supplementary Table 2). Logistic regression for individual pathogen positivity showed that controls had fewer pathogens when age was adjusted for; however, no difference was found for *Cryptosporidium* or *Giardia* (Supplementary Table 2). Other measures of potential microbiological contributors to stunting or epithelial damage (urinary aflatoxin M1 excretion[24] and *Helicobacter pylori* seropositivity[25]) were also sustained or increased with age (Fig. 3b,c). Circulating iFABP concentrations were increased by 0.12 ng ml⁻¹ for every additional enteric pathogen at baseline (β = 0.12; P = 0.02; Extended Data Fig. 4). In multivariable regression analysis, only *Cryptosporidium* infection was correlated with iFABP concentrations in the whole group (β = 1.35;

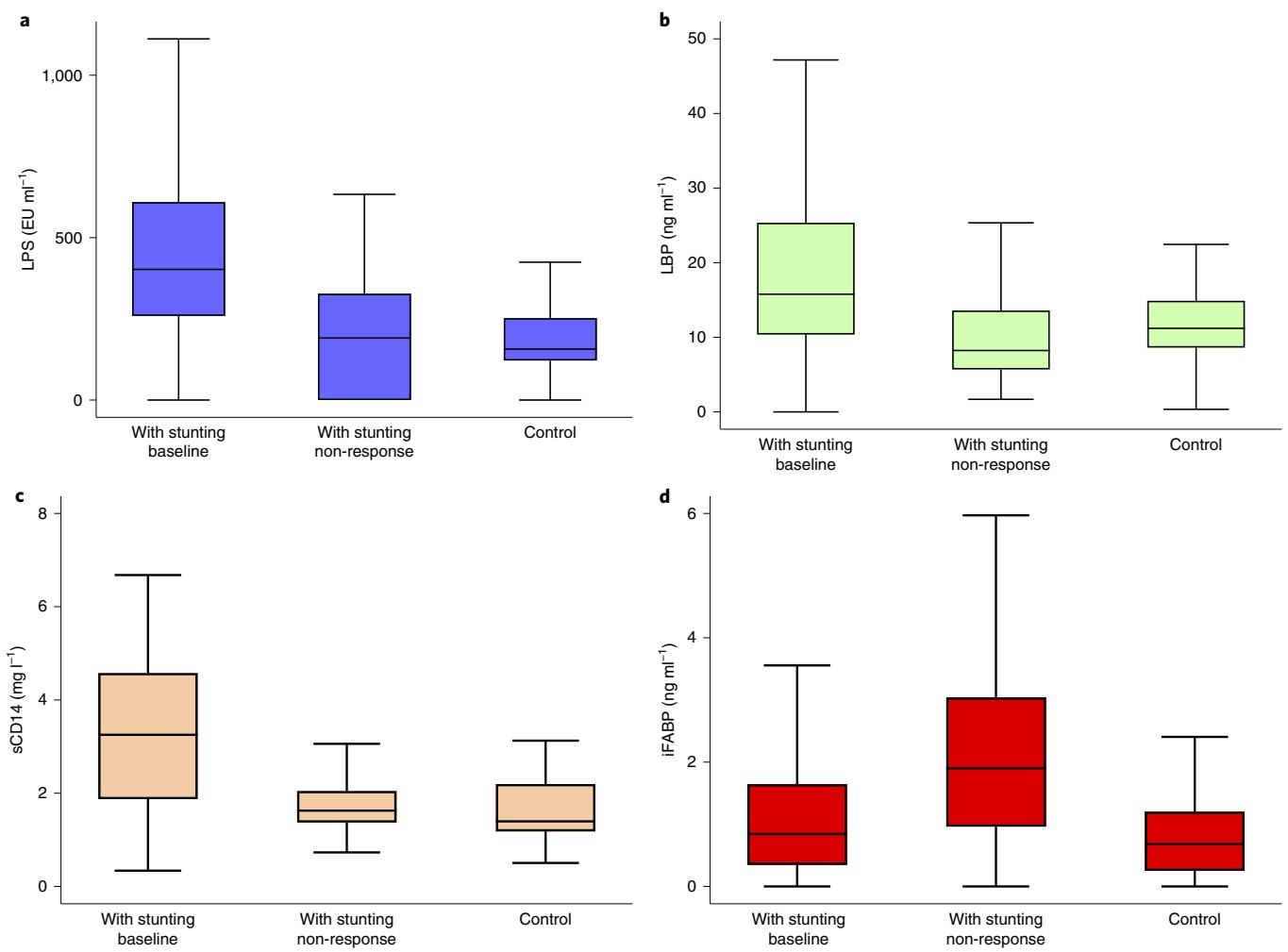

**Fig. 1 | Biomarkers of microbial translocation in children with stunting and controls.** Plasma concentrations of biomarkers in children with stunting at baseline ($n=297$), when non-response was declared ($n=108$) and in controls at baseline ($n=44$). **a–d**, LPS (**a**); LBP (**b**), sCD14 (**c**) and iFABP (**d**). Differences between baseline samples in children with stunting and controls were significant for LPS, LBP and sCD14 ($P=0.0001$ by two-sided Kruskal–Wallis test) but not iFABP. To compare those children who had both baseline samples and samples when non-response was declared, the Wilcoxon signed-rank test was used: LPS ($P<0.001$); LBP ($P=0.006$); sCD14 ($P<0.001$); iFABP ($P=0.01$). Several extreme values have been omitted from the graphs to allow easier visual assessment: 2 LPS values of over 3,000 EU ml⁻¹ at baseline and 3 iFABP values of over 10 ng ml⁻¹, two of which were at baseline. However, these values were included in all statistical analyses. The box plot shows the median, IQR and 5th and 95th centiles.

$P<0.001$), which included 37 children who were reported to have had diarrhoea in the 7 d before sampling. The association remained true in children who had not experienced any diarrhoea in stools within the previous 7 d ($\beta=1.37$; $P=0.002$).

**Endoscopic biopsy and confocal laser endomicroscopy of mucosae.** We performed oesophagogastroduodenoscopy in 118 children with non-responsive stunting and confocal laser endomicroscopy (CLE) in a subset of 75. None of these children had had diarrhoea in the 14 d before endoscopy. The time taken for initial fogging to clear from the confocal endomicroscope lens and the time while the probe tip was not in contact with the mucosa left a mean of 49 s (range 27–106) of assessable CLE video recording for each child. Fluorescein leakage was quantified both as the number of leakage events (plumes) and as the total proportion of imaged time during which luminal fluorescein was visible (Fig. 4); these measures were highly correlated (Extended Data Fig. 5). Interpreters of the videos were blinded to the clinical or research data from individual children. Fluorescein leakage varied from 0 to 100% of video time assessed (median = 29%, IQR = 16–46%). No sex difference was

observed. Fluorescein leak, as a proportion of time imaged, declined with age (regression coefficient $\beta=-0.022$; $P<0.001$) and was inversely correlated with crypt depth ($\rho=-0.34$; $P=0.01$; Fig. 4).

Three duodenal biopsies were taken from each child undergoing oesophagogastroduodenoscopy. Biopsies demonstrated severe enteropathy in these asymptomatic children (Fig. 5). Villus height (median = 180 μm, IQR = 144–216) and crypt depth (median = 171 μm, IQR = 147–203) were similar to measurements previously reported (median = 211 and 157 μm, respectively) in children with severe acute malnutrition in our centre[21]. Biopsies from two children showed total villus atrophy. Villus height, crypt depth and villus surface area, as measured in intestinal biopsies, did not change with age in children with non-responsive stunting.

**Transcriptomic analysis of microbial translocation.** Since intestinal-type alkaline phosphatase (ALPI), expressed in the intestinal microvillus brush border, is known to hydrolyze LPS[26], we sought evidence that specific expression of the *ALPI* gene could explain changes in circulating LPS (Supplementary Table 3) using RNA sequencing on biopsies from 30 children. No significant

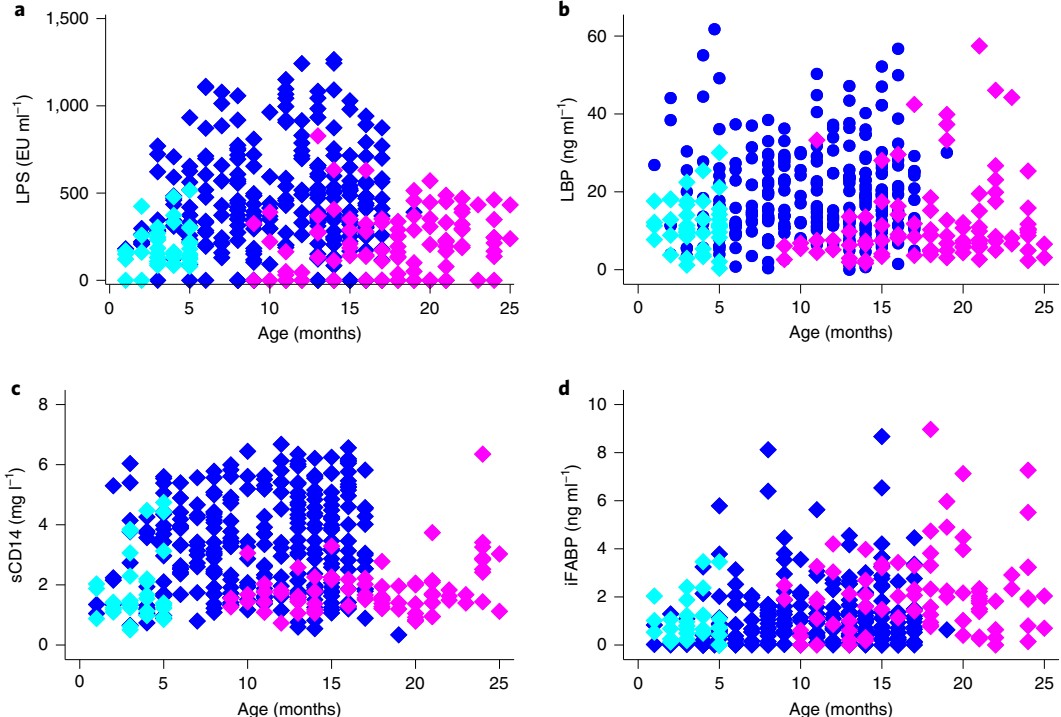

**Fig. 2 | Biomarkers of microbial translocation at different ages. a–c**, Plasma concentrations of LPS (**a**), LBP (**b**), sCD14 (**c**) and iFABP (**d**) in children sampled at different ages, some of whom were sampled twice. Values in children with stunting at baseline ($n = 297$) are shown in blue, when non-response was declared ($n = 108$) in magenta and in controls ($n = 44$) in cyan. Again, a small number of extreme values have been omitted to bring the $y$ axis scale into an appropriate range.

relationship was found between expression and age or between ALPI expression and LPS. However, circulating LPS concentrations were significantly inversely correlated with three other brush border enzymes (Supplementary Table 3): maltase ($\rho = -0.45$; $P = 0.03$); folate hydrolase ($\rho = -0.37$; $P < 0.05$); and angiotensin converting enzyme ($\rho = -0.45$; $P = 0.03$). In general, expression of these enzymes was highly correlated (Extended Data Fig. 6).

To determine if changes in the expression of genes encoding specific barrier functions could explain changes in the microbial translocation observed, we selected 73 transcripts of barrier-relevant genes (that is, those affecting mucus secretion or stability, tight junction function or antimicrobial activity (Supplementary Table 4). None of these transcripts varied significantly with age; therefore, they cannot explain the observed decline in microbial translocation. Circulating LPS concentrations were inversely correlated with the expression of MUC13 and MUC17 ($\rho = -0.38$ and $-0.37$, respectively; $P = 0.04$ for both) and LBP concentrations were inversely associated with MUC4 ($\rho = -0.45$; $P = 0.01$). Circulating sCD14 also inversely varied with claudin-4 ($\rho = -0.46$; $P = 0.01$) expression. Taken together, these correlations emphasize the importance of barrier proteins in controlling microbial translocation but do not explain the inferred decrease in translocation with increasing age.

To determine if absorptive function reflects the adaptive response, we analysed gene expression in a set of 21 key solute carrier transcripts, selected to represent the uptake of key nutrients and on the basis of abundance (Supplementary Table 5). None correlated with age but several inversely correlated with translocation markers and iFABP, as would be expected in damaged epithelium.

**HIV and study population.** At recruitment, 10 and 83 children were HIV-infected or exposed but uninfected, respectively (Supplementary Table 6). LAZ was lower in children who were HIV-exposed but uninfected and lower still in children with

active HIV infection ($P = 0.006$ by non-parametric trend test; Supplementary Table 6). Circulating markers of translocation did not vary significantly with HIV status (Supplementary Table 6). The effect of HIV on confocal endomicroscopy scores, villus height and crypt depth could not be evaluated because, by chance, only one HIV seropositive child underwent endoscopy.

## Discussion

Understanding the development of environmental enteropathy in children living in unsanitary, impoverished environments is of prime importance if we are to devise methods to prevent it. Our findings in Lusaka show that young children with non-responsive stunting have continuous, intense exposure to enteropathogens and remarkably severe ongoing enteropathy, with greatly reduced villus height and villus:crypt ratios barely greater than unity. Three biomarkers of microbial translocation all showed markedly higher levels at baseline in children with stunting but, counter-intuitively, reverted towards control values over the next year or more despite continuing, often severe, growth faltering. Children who did not respond to nutritional supplementation had significantly lower microbial translocation marker levels than 6–12 months previously. Consistent with this, leakage of fluorescein, imaged using CLE, also declined with increasing age. We propose that the pathology we report indicates that environmental enteropathy is an adaptive response that reduces the potentially lethal exposure to microbial translocation that occurs owing to polymicrobial pathogen-induced epithelial damage. Such an adaptation might confer a survival advantage in settings where pathogen-mediated intestinal damage is a substantial burden.

Environmental enteropathy is associated with high, sustained circulating plasma iFABP concentrations indicative of continuing structural damage to the epithelium. Indeed, iFABP concentrations increased over time in children who failed to respond to nutritional

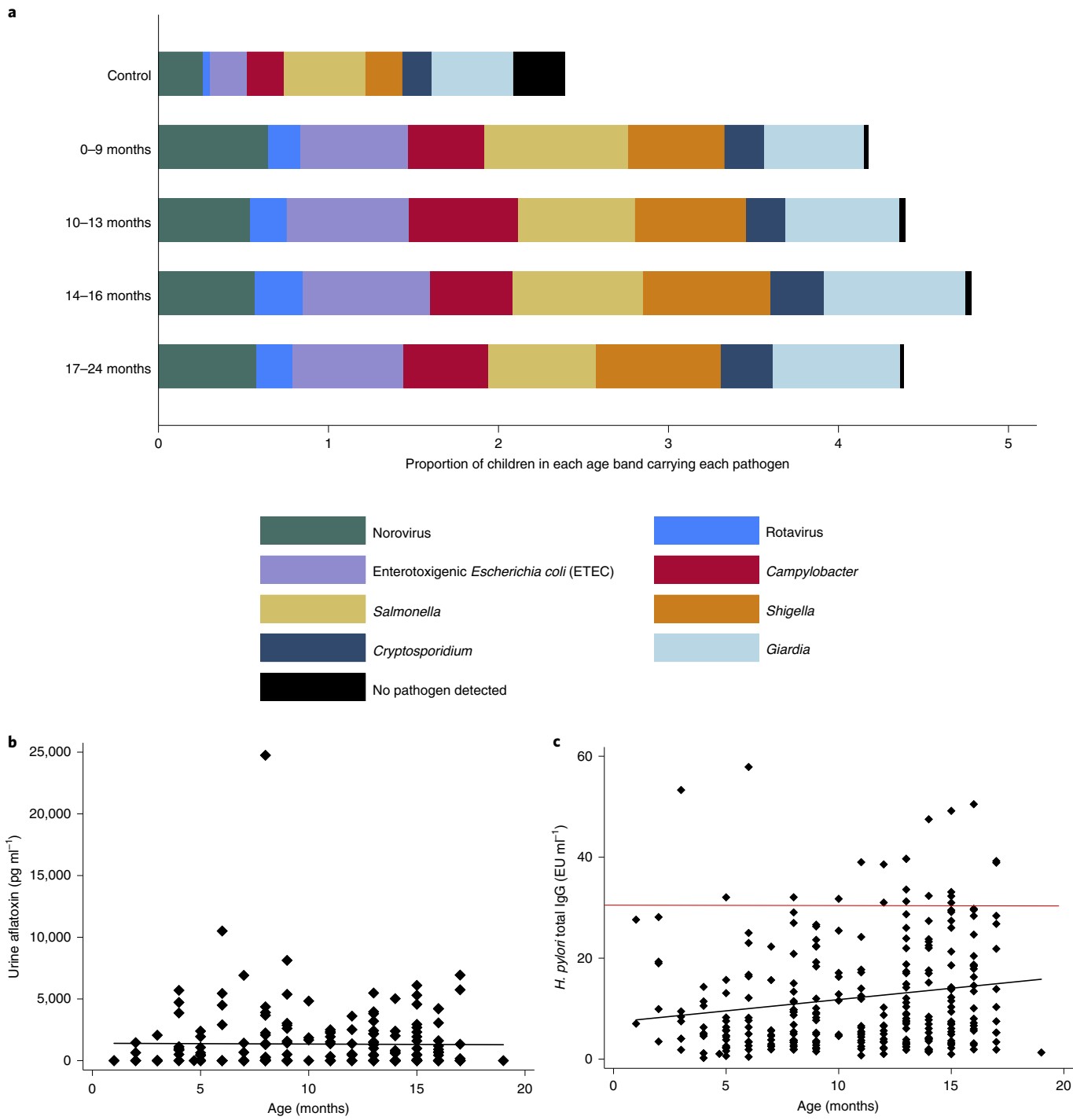

**Fig. 3 | Pathogen burden in children with stunting and controls.** Pathogen burden did not decrease in children with non-responsive stunting over the first 22 months of life. **a**, Proportion of stool samples positive for each pathogen shown in children with stunting in each of four quartiles of age, compared to controls (who were all aged 0–9 months). The proportion of positive stool sample totals was over 4.0 (400%) due to the high number of concurrent multiple pathogens detected in each sample. Some children submitted two samples at different time points. **b**, Continuing or increasing exposure to aflatoxin, measured as aflatoxin M1 (pg ml⁻¹) in urine samples. **c**, Seropositivity to *H. pylori* increased with age, with 4% of children under 9 months of age being seropositive but 11% of children over that age positive. Absolute serological values also increased ($\beta = 0.45$; $P = 0.005$). The red line shows the cut-off for positivity (30 EU ml⁻¹). Regressions were not adjusted for multiple testing.

interventions and circulating iFABP was inversely associated with LAZ. iFABP was positively associated with pathogen burden, especially cryptosporidiosis. Taken together with histological and other evidence, this clearly indicates ongoing epithelial damage associated with enteropathogens and with stunting.

We considered several explanations for reduced epithelial leakage concurrent with unexpectedly reduced biomarkers of microbial translocation in children with non-responsive stunting. Epithelial healing is one candidate but the persistence of elevated iFABP concentrations argues against this. Improved tight junction integrity

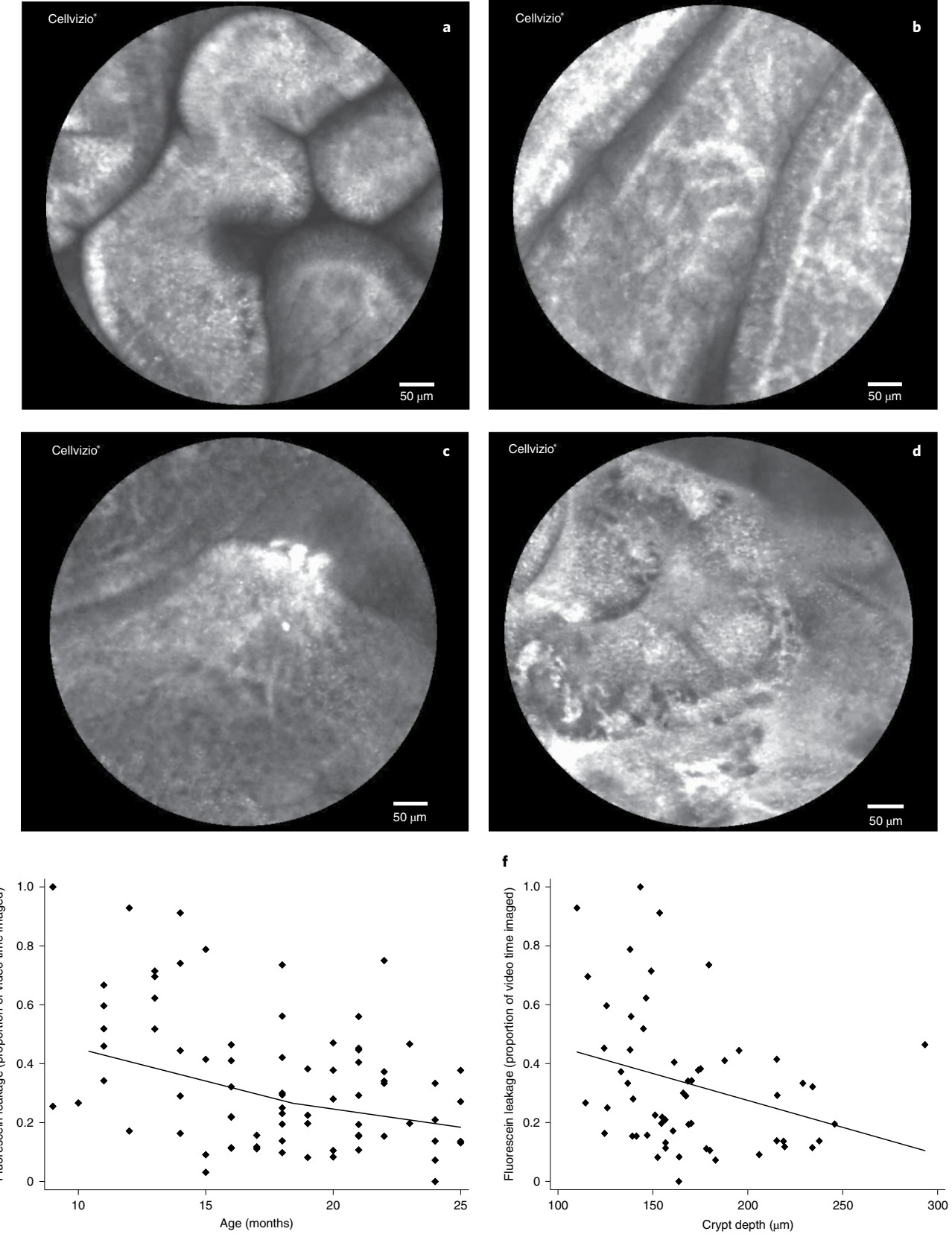

**Fig. 4 | CLE images of barrier impairment. a–d,** Normal epithelium illuminated by fluorescein, with dark lumen (**a**), fluorescein-filled capillaries seen underlying the epithelium (**b**), plumes of fluorescein from an epithelial defect (**c**) and extensive leak of fluorescein from the epithelium (**d**). **e,** Reduction in epithelial barrier dysfunction (as measured by fluorescein leakage) with increasing age over the first 2 years of life in children with non-response ($\beta = -0.022$; $P < 0.0001$). **f,** Fluorescein leak was inversely correlated with crypt depth ($\beta = -0.002$; $P = 0.03$) but not villus height or epithelial surface area (data not shown). Regressions were not adjusted for multiple testing.

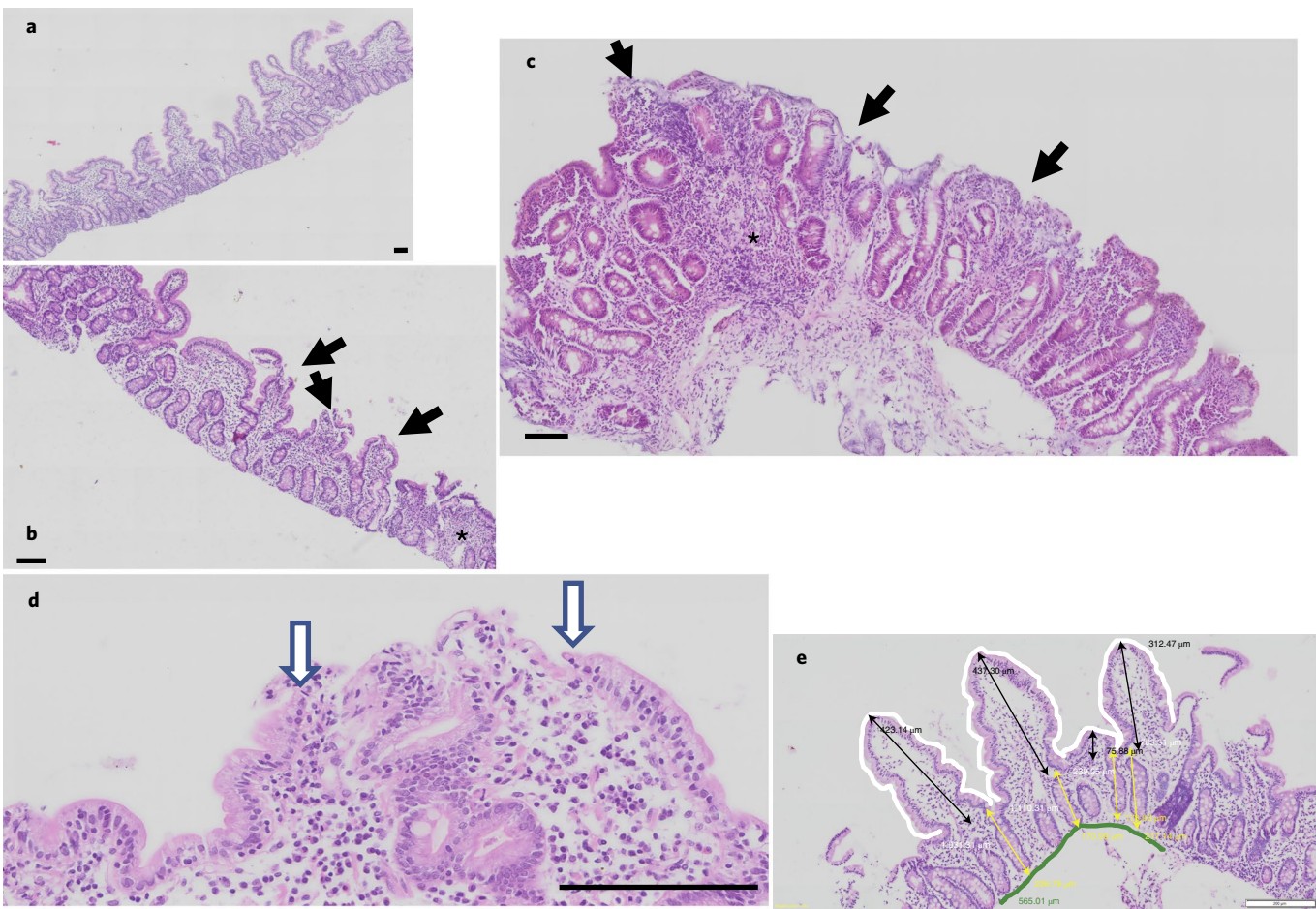

**Fig. 5 | Small intestinal biopsies from children with refractory stunting. a**, Biopsy with tall villi, little inflammation and intact epithelium, but some early villus fusion apparent. **b**, Moderate enteropathy with villus blunting and epithelial damage (solid arrows) and a focus of chronic inflammation (marked by the asterisk). **c**, Severe enteropathy with subtotal villus blunting and areas of epithelial loss (solid arrows) and a focus of chronic inflammation (marked by the asterisk). **d**, Close-up of epithelial damage showing a microerosion characterized by the progressive loss of enterocyte height at the edges, which are shown by the hollow arrows. **e**, Morphometric analysis showing the measurements of villus height (black lines), crypt depth (yellow lines), villus surface area (white perimeter) and length of muscularis mucosae (green line), which is the denominator for calculating the epithelial surface area. **a–d**, Scale bars, 100 μm.

might also explain the observed change but we observed no change in the specific expression of tight junction protein genes with age. Maturation of the microbiota might underlie the apparent adaptation because immaturity of the gut microbiota is associated with undernutrition[27,28] and restoration of microbial communities increases colonization resistance to pathogens[29]. Recent evidence from the adoptive transfer of duodenal microbiota from children with stunting in Bangladesh into gnotobiotic mice identifies its role in the health of the mucosa[30]. Taken together with our data, this seems to suggest that the health of the mucosa depends on the composition, and presumably metabolic activity, of the microbial communities in the small intestinal lumen and mucus layer. We speculate that quorum sensing may mediate the interplay between commensals and pathobionts, which is now an area ripe for further study. One of the great unknowns in microbial translocation is the fractional extraction of pathogen-associated molecular patterns by Kupffer cells in the hepatic sinusoids[31]. An increase in this fractional extraction would explain reduced translocation but this would not explain the reduction in leakage observed by CLE and it is most likely that the adaptive response is located in the intestinal mucosa.

Our data raise the intriguing possibility that reduced absorptive surface area is the reason for diminishing translocation. Such an organ response would reduce overall translocation and fluorescein leakage. The lack of detectable change in villus morphology with age in the biopsy sample is compatible with this process since the small pinch biopsies that can be obtained in young children are unlikely to represent fully changes that occur across the very large surface of the whole small intestine. Biopsies, which often have only 6–20 measurable villi, are not ideal to measure surface area due to this sampling problem. Furthermore, transcriptomic analysis uses a measure of cell mass as its denominator and changes in total surface area may lead to changes in the absolute number of solute transporter or mucin or defensin molecules without a measurable change in the specific expression (fragments per kilobase of transcript per million mapped reads) value. Changes in surface area could also be attributable to changes in the surface area of the microvilli[32], which are difficult to measure. Such an adaptation, reduced total surface area caused by blunting of villi and microvilli, would necessarily incur a cost, that is, reduced nutrient absorptive capacity. In other words, adaptation to enteropathy in children with refractory stunting in Lusaka represents an extensive response of the small bowel, which is not apparent in the intensive assessment of pinch biopsies necessarily limited to very small axial lengths.

Loss of surface area would explain why children with stunting do not respond well to nutrient supplementation and we propose that this is the most parsimonious explanation for our findings.

We recognize that our interpretation of these data is constrained by several important limitations. Most importantly, because we have no ethical justification for endoscopy on children who are healthy, we have almost no information on healthy children of the same ages from the same region. Many mothers of healthy children were also unwilling to participate once told their children were growing well, which explains why a small fraction of the potential controls actually consented. It is also a limitation that morphometry can only be performed on a few very small biopsies from the second or third part of the duodenum; it is difficult to believe that these can fully represent the overall state of the mucosa of the distal duodenum, jejunum and ileum. The carriage rate of enteropathogens, measured by PCR using the xTAG Gastrointestinal Pathogen Panel (GPP), was higher than studies from other settings. However, it must be emphasized that our study was entirely conducted in a very disadvantaged community with poor sanitation and very few resources. We are aware of only one other study in Zambia using the same diagnostic tool[33]; that study included rural areas and a broader range of urban socio-economic conditions and found carriage rates about half of those reported in this study. Molecular diagnosis has been widely used for analysis of enteropathogen burden and the xTAG GPP panel has been reported to deliver accurate pathogen detection[34,35]. The eight-country MAL-ED study demonstrated the usefulness of PCR-based diagnosis[36] but as always the biological significance of low-intensity infections is unclear.

The four biomarkers on which we have focused all report different facets of pathophysiology. iFABP is an intracellular, mainly cytosolic protein of the enterocyte involved in long-chain fatty acid absorption[37], which is released into the bloodstream by epithelial damage and is used as a biomarker of coeliac disease and environmental enteropathy[38]. LPS, a cell wall component of Gram-negative bacteria, is measured directly[39] using the *Limulus* amoebocyte lysate assay or indirectly using host-response molecules such as LBP and sCD14. These host molecules belong to the Toll-like receptor 4 complex on monocytes and macrophages, one of several LPS recognition systems[40]. If it is confirmed that the small intestinal mucosa adapts over time to a hostile environment through reduced microbial translocation, this might explain why some studies show a weak relationship between biomarkers of enteropathy based on microbial translocation and growth failure[41,42].

CLE allows direct imaging of the small intestinal mucosa and fluorescein efflux allows quantification of epithelial leakage in vivo[43,44]. The reduction in fluorescein leakage with age is consistent with the reduction in microbial translocation shown using three blood biomarkers. Fluorescein has similar molecular size to lactulose, a widely used marker of paracellular permeation[45], and is emerging as an alternative way of assessing permeability[46]. We observed multiple fluorescein plumes emitted from the duodenal epithelium, which we interpret as evidence of breaches in epithelial continuity (Figs. 4 and 5). Fluorescein leakage measured endoscopically would reflect loss of epithelial continuity from microerosions but would also be reduced by any reduction in the surface area of damaged enterocytes. We previously reported fluorescein leakage in adults with environmental enteropathy from the same community in Lusaka as the children we report in this study[43,44]. It is of interest that increased crypt depth was associated with reduced fluorescein leakage. This might reflect the proliferative response, since crypt hypertrophy is generally a marker of increased proliferation[47], and therefore greater capacity to replace damaged cells on the villus epithelium.

We propose that environmental enteropathy is an adaptation that sacrifices long-term nutritional health and stature in favour of short-term survival. If so, epithelial healing might emerge as the most essential requirement for catch-up growth. Therapies that chelate pathogen-associated molecular patterns might prove beneficial since they would rapidly reduce mucosal and systemic inflammation.

## Methods

**BEECH study approval and processes.** The study was approved by the University of Zambia Biomedical and Research Committee (ref. 006-02-16, 31 May 2016). Informed written consent was obtained from the parents or primary caregivers of the children and the study was conducted in compliance with the Declaration of Helsinki (version 2008). The community where this study was entirely conducted is Misisi, a densely populated residential area just south of central Lusaka. Houses are constructed of concrete blocks with iron sheet roofing, sanitation is poor (1 pit latrine per 60 residents) and water supply is from communal standpipes. Domestic animals are free to roam around but few householders keep pets so the numbers of chickens, cats and dogs are low, but rats are frequent pests. In general, consumption of animal source foods is infrequent, but a daily egg was provided by the study team for each participating child.

Children were screened in the community using only weight measurements because length measurements in the community are impractical. Children were monitored every two weeks from recruitment to discharge but length/height was measured only monthly. Non-response was defined as failure to achieve a positive gradient in LAZ over at least 4 months, together with LAZ consistently below −2. In two cases, children booked for endoscopy showed LAZ greater than −2 on the last assessment but endoscopy carried out as part of the overall profile of LAZ was consistently non-responsive until that isolated measurement. Controls ($n = 46$) were simultaneously recruited by identifying children with good growth (LAZ, WAZ and WLZ all greater than −2 at recruitment, but preferably with z-scores all greater than −1) from the same community (Table 1). These children were provided with supplementary food for compassionate reasons but did not undergo endoscopy. In September 2016, recruitment began with inclusion restricted to children under 6 months of age; after January 2017, inclusion criteria were broadened to include undernourished children up to 18 months of age. Recruitment of controls was unchanged at up to six months of age. Recruitment of controls was only initiated after that change had been made. Thus, the final dataset included undernourished children recruited at 0–18 months of age and controls recruited at 0–6 months of age. The period of observation encompassed all seasons of the year since environmental enteropathy is a seasonal disorder[16]. Pathogen analysis was conducted within three months of recruitment. Biomarkers were measured within three months of recruitment and again when non-response was declared.

**Growth and anthropometry.** Children were fully assessed for growth faltering every month by trained study nurses who performed anthropometry in triplicate. Measurements were immediately entered into the WHO Anthro v3.2.2 software for calculation of precise WLZ, LAZ and WAZ scores and checked for plausibility and consistency by a study paediatrician (BA or KC). The instruments used included infant and mobile scales (seca 384 and 874) for weight, infantometer (seca 416) and UNICEF stadiometer for length.

**Pathogen analysis.** Stool samples were collected from cases at baseline and after 3 months of nutritional supplementation, and at baseline only for the controls, and stored (−80 °C) before enteropathogen analysis using the xTAG GPP kit (Luminex Corporation), which qualitatively detects viral, parasitic and bacterial nucleic acids in stool samples. A total of 286 Luminex stool results from cases and controls were available for this analysis; 198 were baseline samples and 88 were repeat samples collected approximately 3 months later.

**Biomarkers of enteropathy and microbial translocation.** Blood samples were taken from children with stunting and from controls at recruitment and in children with stunting with non-response to the nutritional intervention at the time of endoscopy. iFABP is a protein constitutively expressed in intestinal epithelial cells. It is released by cell damage and circulating concentrations reflect the severity of epithelial damage[48,49]. It was measured in plasma by enzyme-linked immunosorbent assay (ELISA; Cambridge Bioscience). LPS, LBP and sCD14 were used as markers of microbial translocation resulting from impairment of the gut barrier function. For LPS analysis, the pyrochrome *Limulus* amoebocyte lysate assay (Associates of Cape Cod) kit was used. Human LBP and sCD14 were assayed by ELISA (R&D Systems). *Helicobacter* serology was evaluated in serum using total antibodies to *H. pylori* by ELISA (BIOHIT HealthCare) and aflatoxin M1 was measured in urine, also by ELISA (Helica Biosystems).

**Endoscopy and CLE.** Once children were declared to have non-response, they were booked to have endoscopy with a Pentax EG-2490k paediatric gastroscope under ketamine-based sedation administered by an anaesthetist. Before endoscopy, several checks were carried out to identify conditions that might predispose to adverse events, including full blood count and measurement of prothrombin time (as international normalized ratio). During endoscopy, an injection of 2 ml 1% fluorescein in normal saline was administered, followed by a 60-s video recording using a Cellvizio confocal endomicroscopy probe (Mauna Kea Technologies). After clearing of initial lens fogging, video was recorded continuously. Endoscopic biopsies were obtained from the second part of the duodenum using standard disposable biopsy forceps and three biopsies were placed in normal saline to facilitate orientation under a dissecting microscope, followed by immersion in

formal saline on cellulose acetate strips (0.45-µm pore; catalogue no. 11106-30-N, Sartorius AG). The maximum period before fixation was 10 min, although usually less. Biopsies were embedded in paraffin wax and stained with haematoxylin and eosin, then imaged on an Olympus VS120 scanning microscope. Morphometry was performed as described previously[44] (Fig. 5).

**RNA sequencing.** RNA sequencing (RNA-seq) was performed on biopsies from the first 30 consecutive children to be declared to have non-responsive stunting, without selection. Two small intestinal biopsies were immediately snap-frozen in liquid nitrogen before storage at −80 °C. RNA was extracted using TRIzol (Invitrogen) followed by silica column purification (RNeasy Mini Kit; QIAGEN) and quantified using a NanoDrop spectrophotometer (ND-2000C; Thermo Fisher Scientific) before transport to the Beijing Genomics Institute. RNA quality control was performed using an Agilent 2100 Bioanalyzer and ABI StepOnePlus Real-Time PCR System. For RNA-seq preparation, total RNA was treated with DNase I followed by messenger RNA enrichment using oligo-deoxythymine-labelled beads and ligation of sequencing adaptors to the enriched mRNA fragments. RNA-seq was carried out using an Illumina HiSeq 2000 Sequencing System with 50 base pair reads. Filtering steps included removing reads with adaptors, removing reads where unknown bases were >10% and removing low-quality reads (percentage of low-quality bases >50%). The proportion of clean reads was never <99% and usually >99.6%. After low-quality reads were removed and adaptor sequences trimmed, 3.3 billion clean reads were obtained, with an average of 165 million reads per sample. mRNA tissue content was expressed as fragments per kilobase mapped values and therefore indicates gene expression per unit mass of tissue sequenced.

**Data analysis.** Biomarkers were assessed at recruitment (children with stunting and controls) and then once non-response was declared (children with stunting only). Biomarkers were generally non-normally distributed so are presented as the median and IQR. Statistical testing was performed using the Kruskal–Wallis test for children with stunting–controls comparisons, Wilcoxon signed-rank test for repeated tests (baseline compared to non-response in children with stunting) and Fisher's exact test for categorical variables; all tests were two-sided. Correlations used Spearman's non-parametric correlation coefficient ($\rho$) and linear regression was used to assess the dependence of selected variables on age (the regression coefficient $\beta$ is quoted where significant at $P < 0.05$). No correction for multiple hypothesis testing was carried out for gene correlations; it was clear that few of these correlations (Supplementary Tables 3–5) were significant and these are listed mainly to demonstrate that adaptation does not appear to operate at the level of specific gene expression. In some graphs, extreme values have been omitted merely to bring data into a more visually acceptable scale; no data were omitted in any analysis.

Pathogens were compared in children with stunting and controls and analysed by quartile of age in children with stunting, as well as total the number of pathogens present in the stool sample from each child. Linear regression of total pathogen numbers and logistic regression of individual pathogen carriage were analysed in a subset of samples restricted to the first sample collected to confirm the effect of age and control status on pathogen carriage (Supplementary Table 2). The relationship between infecting pathogens and circulating iFABP was examined in a backwards stepwise linear regression model with all pathogens with a frequency greater than $n = 5$ included as the independent variables and iFABP (raw and square root-transformed) as the dependent variable; the statistical associations were identical, so the regressions of raw iFABP are reported. Analysis of endomicroscopy videos was performed in two ways. First, by assessing each still frame (10 per second) for the presence of fluorescein leakage (Fig. 1). We have previously reported that fluorescein leakage is the most reliable (lowest inter-observer variation) measure of epithelial damage[44]. Images with movement artefact and images where the epithelium could not be clearly visualized were not evaluated. Images with fluorescein leakage were expressed as a proportion of all images evaluated. Second, continuous play of each video recording was viewed to measure the proportion of time recorded during which fluorescein was seen to be leaking through the epithelium. These two measurement methods, evaluated by different observers, were moderately correlated ($\rho = 0.44$; $P = 0.0001$; Extended Data Fig. 5). Both measures were used in all analyses and similar results were obtained; however, only the second is reported in the Results for economy of space.

**Reporting Summary.** Further information on research design is available in the Nature Research Reporting Summary linked to this article.

## Data availability

Data have been deposited with Dryad (doi:10.5061/dryad.zkh18937z; https://datadryad.org/stash/share/fYdcG3Ao8fnLwUbwF9N4RIJ4BYfvToSsg-iYuydWK34). This project contains the following data: data file 1—CSV file of pathogens in stool samples from children with stunting and their controls; data file 2—CSV file of biomarkers from children with stunting and their controls. Data are available under the terms of the Creative Commons Zero 'No Rights Reserved' data waiver (CC0 1.0 Public Domain Dedication). Transcriptomic data have been deposited with the Gene Expression Omnibus under accession no. 162630.

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

## Acknowledgements

We thank our endoscopy nurses and anaesthetists for assistance with the endoscopy procedures and to R. Chilengi and staff at the Centre for Infectious Disease Research in Zambia laboratory for work on the xTAG pathogen analysis. We thank R. Feldman, I. Sanderson, P. Tarr, D. Denno, P. Sullivan and K. Jones for helpful discussions. We thank T. Wylie, N. Shaikh and M. Ndao for invaluable technical support. This work was supported by the Bill & Melinda Gates Foundation, through the Environmental Enteric Dysfunction Consortium (grant no. OPP 1066118).

## Author contributions

B.A. and P.K. designed the study. B.A., K. Chandwe, R.B. and K. Chifunda designed the data collection processes, clinical follow-up procedures and data collection. K.Z., E.B., C.M., S.M. and S.S. performed the biomarker analyses. L.K., S.C., M.C., K.Z., K. Chandwe and K.V. were responsible for data management. P.K. and K.V. analysed the data and P.K. wrote the first draft of the manuscript. All authors contributed to checking and revising later versions of the manuscript.

## Competing interests

The authors declare no competing interests.

## Additional information

**Extended data** is available for this paper at https://doi.org/10.1038/s41564-020-00849-w.

**Correspondence and requests for materials** should be addressed to P.K.

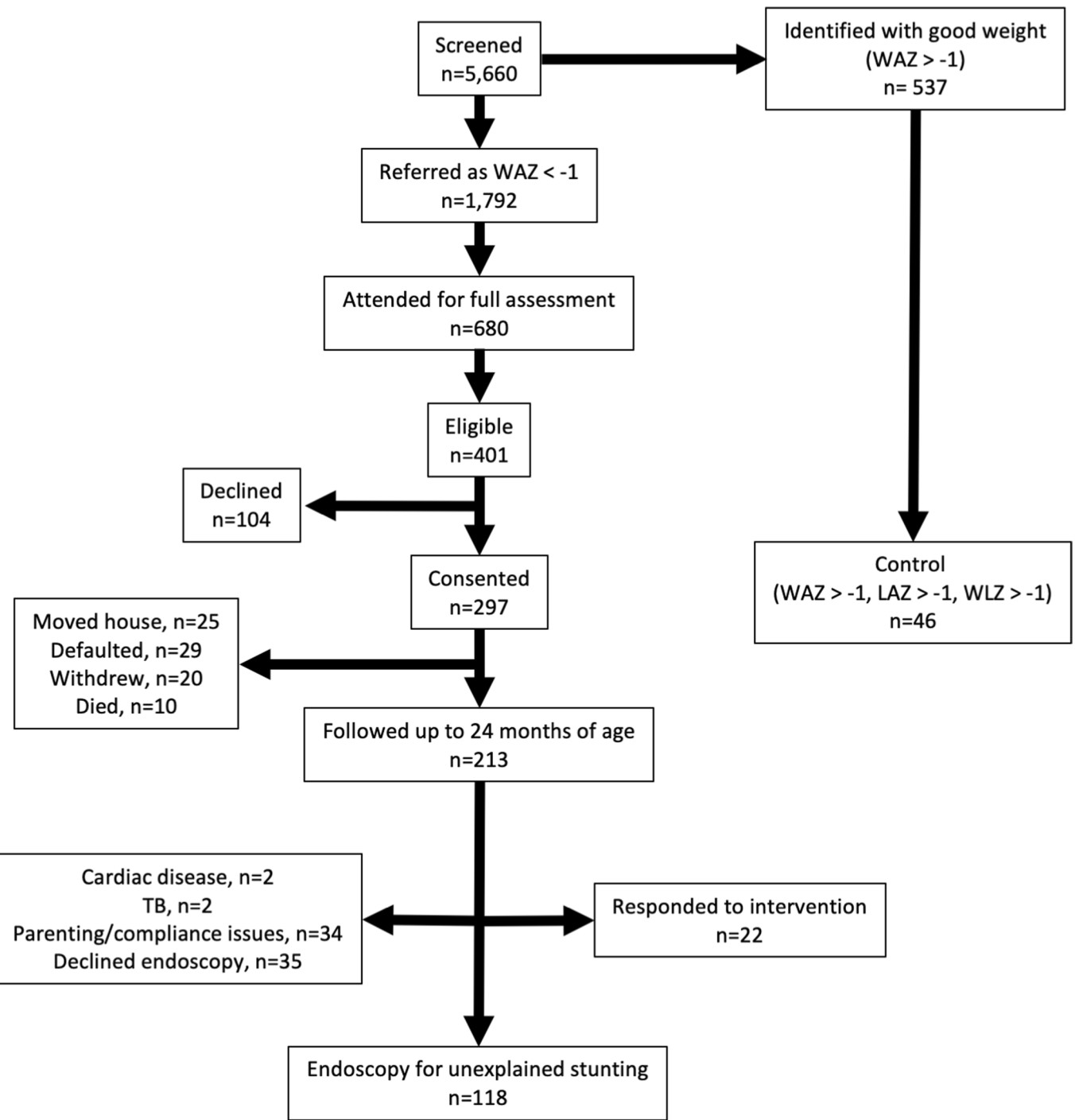

**Extended Data Fig. 1 | Flowchart of screening, recruitment and follow-up.** In September 2016, recruitment began with inclusion restricted to children under 6 months of age, but after January 2017 inclusion criteria were broadened to include undernourished children up to 18 months of age; recruitment of controls remained unchanged at up to 6 months of age. Recruitment of controls was only initiated after that change had been made. Thus the final dataset included undernourished children recruited at 0–18 months of age and controls recruited at 0–6 months of age. From March 2017, recruitment of cases and controls proceeded simultaneously, encompassing 2 full annual seasonal cycles. Pathogen analysis was conducted within 3 months of recruitment. Biomarkers were measured within 3 months of recruitment and again when non-response was declared.

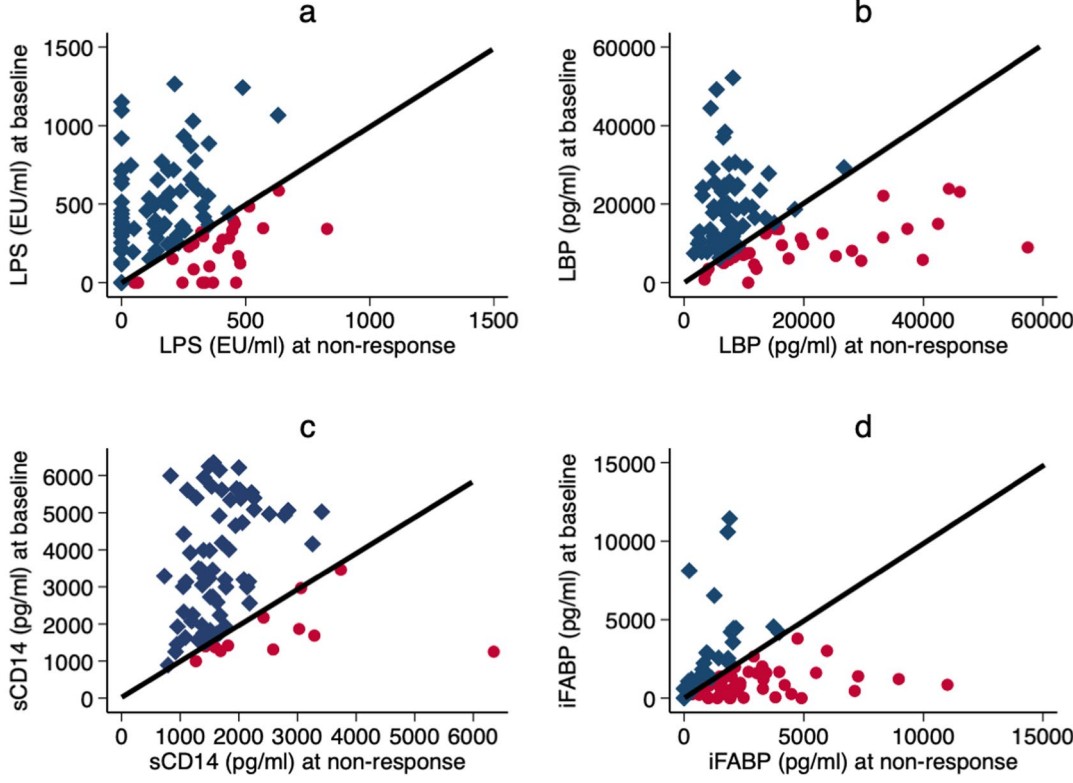

**Extended Data Fig. 2 | Biomarkers at baseline and non-response.** Scatter plots of baseline values of biomarkers against values at non-response in the same child. The black line shown is the line of identity, so points above and to the left (coloured in navy) represent children whose baseline values were higher than when non-response was declared. Conversely, points below and to the right of the line denote children whose values had increased by the date of non-response. For the biomarkers of microbial translocation (LPS, LBP and sCD14) the proportions which showed a decrease were 75%, 64% and 86% respectively, whereas iFABP increased in 63%. LPS, lipopolysaccharide; LBP, LPS binding protein; sCD14, soluble CD14; iFABP, intestinal fatty acid binding protein.

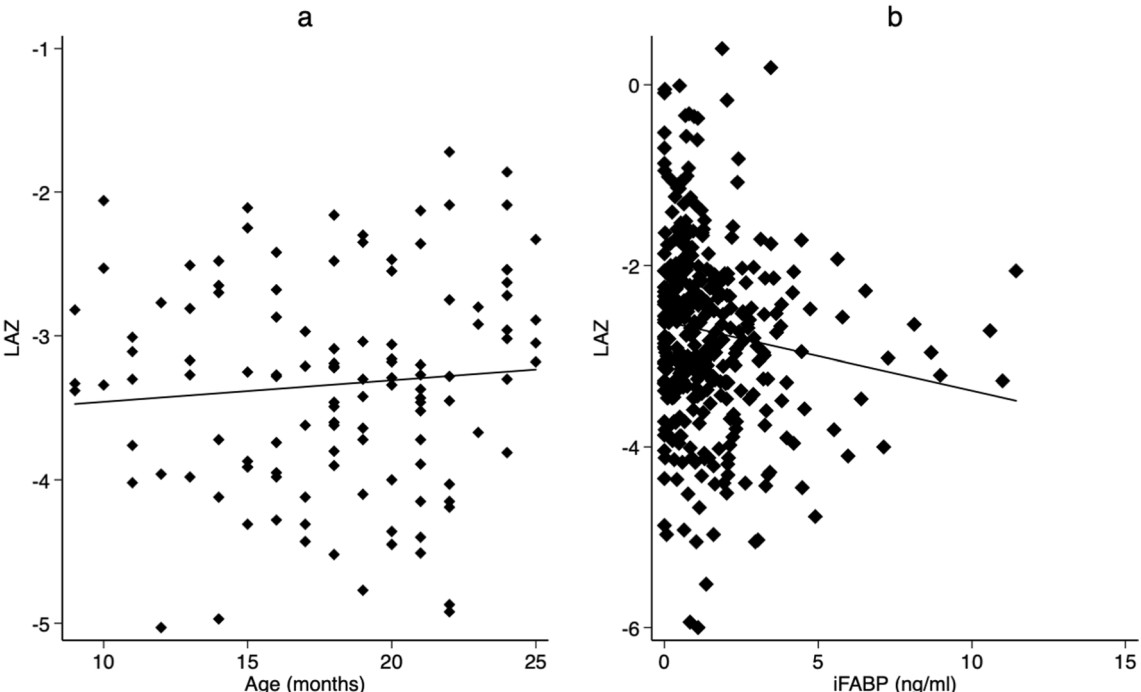

**Extended Data Fig. 3 | Length for age z score in relation to age and epithelial damage. a**, Absence of regression of LAZ on age; the regression line is shown but was not significant ($P = 0.36$). **b**, Intestinal fatty acid binding protein (iFABP) was inversely correlated with LAZ score at the time when endoscopy was performed ($\rho = -0.15$; $P = 0.003$). Using linear regression to estimate the dependence of LAZ on iFABP, despite the non-normal distribution of iFABP, each unit ng/ml increase in FABP was associated with 0.08 z score reduction in LAZ ($\beta = -0.08$; $P = 0.01$). None of the biomarkers of microbial translocation correlated with LAZ.

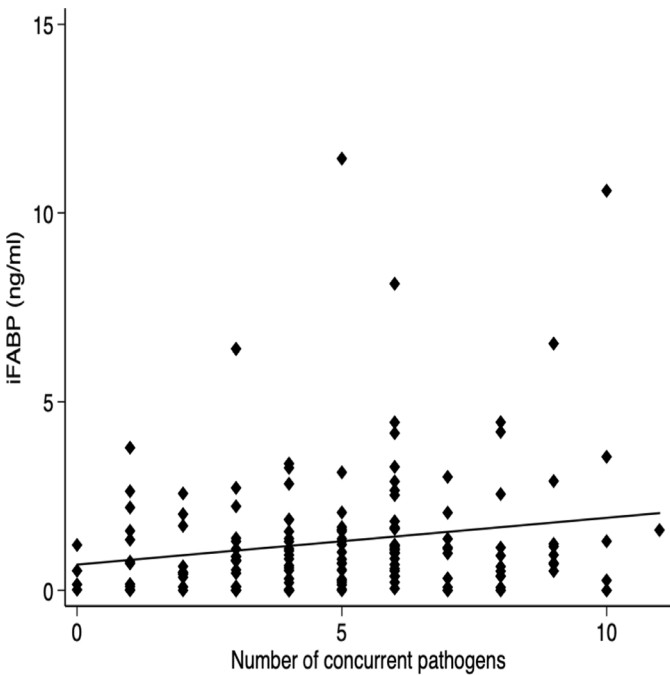

**Extended Data Fig. 4 | Relationship between iFABP and number of pathogens detected concurrently.** Intestinal Fatty Acid Binding Protein (iFABP) was related with the total number of pathogens carried at points in time when measurements were made (regression coefficient $\beta = 0.12$; $P = 0.02$).

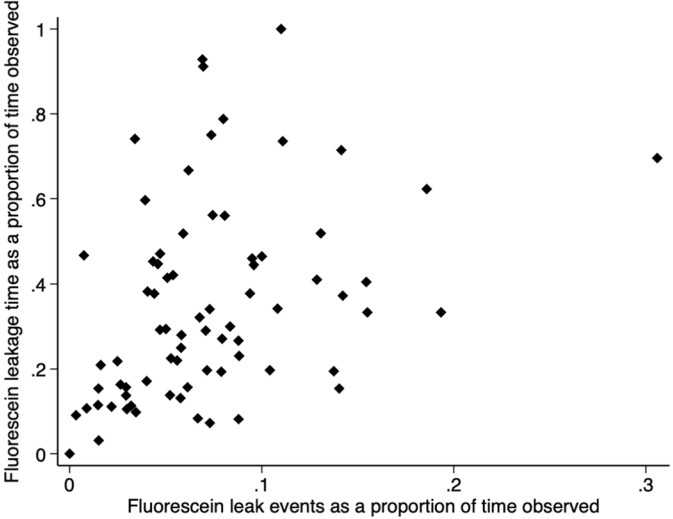

**Extended Data Fig. 5 | Confocal laser endomicroscopy imaging validation.** Concordance between confocal laser endomicroscopy leakage of fluorescein from systemic circulation into gut lumen, assessed using either incidence leakage events or proportion of time imaged.

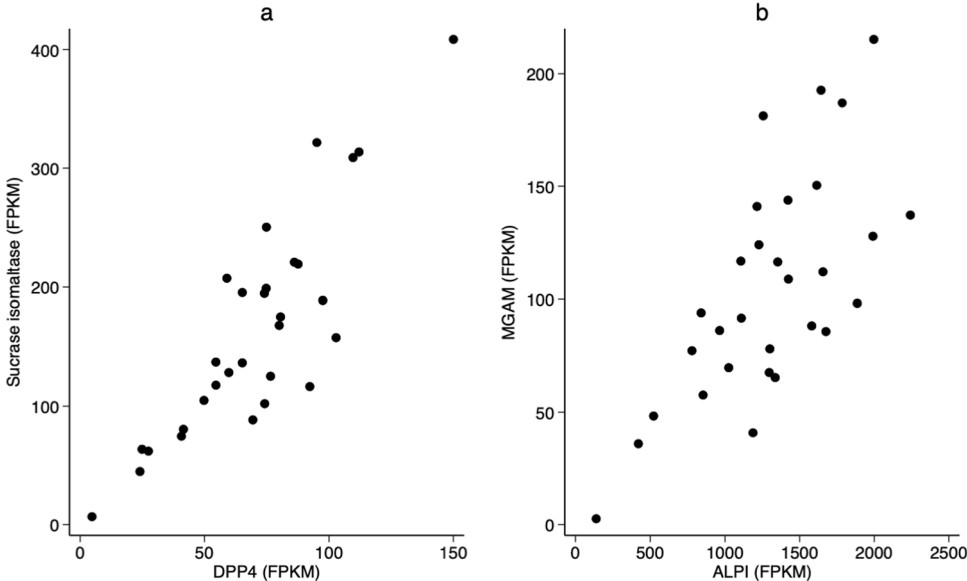

**Extended Data Fig. 6 | Brush border enzyme gene expression.** Brush border enzymes were strongly correlated: examples shown are (**a**) sucrase isomaltase and dipeptidylpeptidase IV (DPP4; $\rho = 0.78$; $P < 0.0001$); **b**, maltase glucoamylase (MGAM) and alkaline phosphatase (ALPI; $\rho = 0.65$; $P = 0.0001$).

# nature research

# Reporting Summary

Nature Research wishes to improve the reproducibility of the work that we publish. This form provides structure for consistency and transparency in reporting. For further information on Nature Research policies, see Authors & Referees and the Editorial Policy Checklist.

## Statistics

For all statistical analyses, confirm that the following items are present in the figure legend, table legend, main text, or Methods section.

| n/a | Confirmed | |
|---|---|---|
| ☐ | ☒ | The exact sample size (*n*) for each experimental group/condition, given as a discrete number and unit of measurement |
| ☐ | ☒ | A statement on whether measurements were taken from distinct samples or whether the same sample was measured repeatedly |
| ☐ | ☒ | The statistical test(s) used AND whether they are one- or two-sided *Only common tests should be described solely by name; describe more complex techniques in the Methods section.* |
| ☐ | ☒ | A description of all covariates tested |
| ☐ | ☒ | A description of any assumptions or corrections, such as tests of normality and adjustment for multiple comparisons |
| ☐ | ☒ | A full description of the statistical parameters including central tendency (e.g. means) or other basic estimates (e.g. regression coefficient) AND variation (e.g. standard deviation) or associated estimates of uncertainty (e.g. confidence intervals) |
| ☐ | ☒ | For null hypothesis testing, the test statistic (e.g. *F*, *t*, *r*) with confidence intervals, effect sizes, degrees of freedom and *P* value noted *Give P values as exact values whenever suitable.* |
| ☒ | ☐ | For Bayesian analysis, information on the choice of priors and Markov chain Monte Carlo settings |
| ☒ | ☐ | For hierarchical and complex designs, identification of the appropriate level for tests and full reporting of outcomes |
| ☐ | ☒ | Estimates of effect sizes (e.g. Cohen's *d*, Pearson's *r*), indicating how they were calculated |

*Our web collection on statistics for biologists contains articles on many of the points above.*

## Software and code

Policy information about availability of computer code

| Data collection | *n/a* |
|---|---|
| Data analysis | *n/a* |

For manuscripts utilizing custom algorithms or software that are central to the research but not yet described in published literature, software must be made available to editors/reviewers. We strongly encourage code deposition in a community repository (e.g. GitHub). See the Nature Research guidelines for submitting code & software for further information.

## Data

Policy information about availability of data

All manuscripts must include a data availability statement. This statement should provide the following information, where applicable:
- Accession codes, unique identifiers, or web links for publicly available datasets
- A list of figures that have associated raw data
- A description of any restrictions on data availability

# Field-specific reporting

Please select the one below that is the best fit for your research. If you are not sure, read the appropriate sections before making your selection.

☒ Life sciences    ☐ Behavioural & social sciences    ☐ Ecological, evolutionary & environmental sciences

# Life sciences study design

All studies must disclose on these points even when the disclosure is negative.

| | |
|---|---|
| Sample size | No formal sample size calculation was performed, as this was part of a collaborative exploratory study. The sample size was based on a previous studies in which (a) transcriptomic analysis, and (b) mucosal morphometry, had proved informative |
| Data exclusions | No data were excluded |
| Replication | All assays were performed in duplicate. Confocal laser endomicroscopy videos were scored by two observers and their agreement is shown in Supplementary Figure 5. |
| Randomization | N/A |
| | N/A |

# Reporting for specific materials, systems and methods

We require information from authors about some types of materials, experimental systems and methods used in many studies. Here, indicate whether each material, system or method listed is relevant to your study. If you are not sure if a list item applies to your research, read the appropriate section before selecting a response.

## Materials & experimental systems

| n/a | Involved in the study |
|---|---|
| ☒ | Antibodies |
| ☒ | Eukaryotic cell lines |
| ☒ | Palaeontology |
| ☒ | Animals and other organisms |
| ☐ | ☒ Human research participants |
| ☐ | ☒ Clinical data |

## Methods

| n/a | Involved in the study |
|---|---|
| ☒ | ChIP-seq |
| ☒ | Flow cytometry |
| ☒ | MRI-based neuroimaging |

## Human research participants

Policy information about studies involving human research participants

| | |
|---|---|
| Population characteristics | Children living in Lusaka, Zambia, with linear growth faltering |
| Recruitment | Door-to-door screening |
| Ethics oversight | The study protocol was approved by the University of Zambia Biomedical Research Ethics Committee; full details are in the manuscript |

Note that full information on the approval of the study protocol must also be provided in the manuscript.

## Clinical data

Policy information about clinical studies

All manuscripts should comply with the ICMJE guidelines for publication of clinical research and a completed CONSORT checklist must be included with all submissions.

| | |
|---|---|
| Clinical trial registration | N/A |
| Study protocol | This was not a clinical trial, though it was a clinical study |
| Data collection | Lusaka, Misisi residential area, from 1st Sept 2016 - 30th June 2019 |
| Outcomes | This was not a clinical trial |

