## [Peer Review File · Nature Microbiology]

Peer Review Information

Journal: Nature Microbiology

Manuscript Title: Adaptation of the small intestine to microbial enteropathogens in
Zambian children with stunting

Corresponding author name(s): Paul Kelly

Editorial Notes:

Redactions – transferred manuscripts (mention of the other journal) This manuscript has been previously reviewed at another journal. This document only contains reviewer comments, rebuttal and decision letters for versions considered at Nature **Microbiology**. Mentions of the other journal have been redacted.

Reviewer Comments & Decisions:

Decision Letter, initial version:
--

Dear Paul,

Thank you for your patience while your manuscript "Adaptation of the small intestinal epithelium to intestinal pathogens in Zambian children with stunting" was under peer-review at Nature Microbiology. It has now been seen by 4 referees, whose expertise and comments you will find at the of this email. You will see from their comments below that while they find your work of interest, some important points are raised. We are very interested in the possibility of publishing your study in Nature Microbiology, but would like to consider your response to these concerns in the form of a revised manuscript before we make a final decision on publication.

In particular, you will see that most of the referees are happy with your revisions and have no further comments, however, referee #2 makes a couple of points that we will need you to respond to including the statistical approaches used for some analyses and improved clarity in the text around how the data were handled. The rest referees' reports are clear and the remaining issues should be straightforward to address.

If you have not done so already please begin to revise your manuscript so that it conforms to our Article format instructions at <http://www.nature.com/nmicrobiol/info/final-submission/>

The usual length limit for a Nature Microbiology Article is six display items (figures or tables) and 3,000 words. We have some flexibility, and can allow a revised manuscript at 3,500 words, but please consider this a firm upper limit. There is a trade-off of ~250 words per display item, so if you need more space, you could move a Figure or Table to Supplementary Information.

Some reduction could be achieved by focusing any introductory material and moving it to the start of your opening 'bold' paragraph, whose function is to outline the background to your work, describe in a sentence your new observations, and explain your main conclusions. The discussion should also be limited. Methods should be described in a separate section following the discussion, we do not place a word limit on Methods.

Nature Microbiology titles should give a sense of the main new findings of a manuscript, and should not contain punctuation. Please keep in mind that we strongly discourage active verbs in titles, and that they should ideally fit within 90 characters each (including spaces).

Please include a data availability statement as a separate section after Methods but before references, under the heading "Data Availability". This section should inform readers about the availability of the data used to support the conclusions of your study. This information includes accession codes to public repositories (data banks for protein, DNA or RNA sequences, microarray, proteomics data etc...), references to source data published alongside the paper, unique identifiers such as URLs to data repository entries, or data set DOIs, and any other statement about data availability. At a minimum, you should include the following statement: "The data that support the findings of this study are available from the corresponding author upon request", mentioning any restrictions on availability. If DOIs are provided, we also strongly encourage including these in the Reference list (authors, title, publisher (repository name), identifier, year). For more guidance on how to write this section please see:

<http://www.nature.com/authors/policies/data/data-availability-statements-data-citations.pdf>

To improve the accessibility of your paper to readers from other research areas, please pay particular attention to the wording of the paper's opening bold paragraph, which serves both as an introduction and as a brief, non-technical summary in about 150 words. If, however, you require one or two extra sentences to explain your work clearly, please include them even if the paragraph is over-length as a result. The opening paragraph should not contain references. Because scientists from other sub-disciplines will be interested in your results and their implications, it is important to explain essential but specialised terms concisely. We suggest you show your summary paragraph to colleagues in other fields to uncover any problematic concepts.

If your paper is accepted for publication, we will edit your display items electronically so they conform to our house style and will reproduce clearly in print. If necessary, we will re-size figures to fit single or double column width. If your figures contain several parts, the parts should form a neat rectangle when assembled. Choosing the right electronic format at this stage will speed up the processing of your paper and give the best possible results in print. We would like the figures to be supplied as vector files - EPS, PDF, AI or postscript (PS) file formats (not raster or bitmap files), preferably generated with vector-graphics software (Adobe Illustrator for example). Please try to ensure that all figures are non-flattened and fully editable. All images should be at least 300 dpi resolution (when figures are scaled to approximately the size that they are to be printed at) and in RGB colour format. Please do not submit Jpeg or flattened TIFF files. Please see also 'Guidelines for Electronic Submission of Figures' at the end of this letter for further detail.

Figure legends must provide a brief description of the figure and the symbols used, within 350 words, including definitions of any error bars employed in the figures.

When submitting the revised version of your manuscript, please pay close attention to our [href="https://www.nature.com/nature-research/editorial-policies/image-integrity">Digital Image Integrity Guidelines.](https://www.nature.com/nature-research/editorial-policies/image-integrity) and to the following points below:

Please include a statement before the acknowledgements naming the author to whom correspondence and requests for materials should be addressed.

Finally, we require authors to include a statement of their individual contributions to the paper -- such as experimental work, project planning, data analysis, etc. -- immediately after the acknowledgements. The statement should be short, and refer to authors by their initials. For details please see the Authorship section of our joint Editorial policies at http://www.nature.com/authors/editorial_policies/authorship.html

* include a point-by-point response to any editorial suggestions and to our referees. Please include your response to the editorial suggestions in your cover letter, and please upload your response to the referees as a separate document.

* ensure it complies with our format requirements for Letters as set out in our guide to authors at www.nature.com/nmicrobiol/info/gta/

* state in a cover note the length of the text, methods and legends; the number of references; number and estimated final size of figures and tables

* resubmit electronically if possible using the link below to access your home page:

{REDACTED]

*This url links to your confidential homepage and associated information about manuscripts you may have submitted or be reviewing for us. If you wish to forward this e-mail to co-authors, please delete this link to your homepage first.

Please ensure that all correspondence is marked with your Nature Microbiology reference number in the subject line.

Nature Microbiology is committed to improving transparency in authorship. As part of our efforts in this direction, we are now requesting that all authors identified as 'corresponding author' on published papers create and link their Open Researcher and Contributor Identifier (ORCID) with their account on the Manuscript Tracking System (MTS), prior to acceptance. This applies to primary research papers only. ORCID helps the scientific community achieve unambiguous attribution of all scholarly contributions. You can create and link your ORCID from the home page of the MTS by clicking on 'Modify my Springer Nature account'. For more information please visit www.springernature.com/orcid.

We hope to receive your revised paper within three weeks. If you cannot send it within this time, please let us know.

Reviewer Expertise:

Referee #1: childhood malnutrition; microbiome

Referee #2: paediatric gastroenterology; sub-Saharan Africa

Referee #3: molecular epidemiology; undernutrition; biomarkers

Referee #4: enteric infections; clinical; nutrition; intestinal barrier dysfunction

Reviewers Comments:

Reviewer #1 (Remarks to the Author):

The authors have responded to my comments with appropriate edits.

Reviewer #2 (Remarks to the Author):

This manuscript reports the results of a sub-study of the Biomarkers of Environmental Enteropathy in Children (BEECH) prospective nutritional intervention study conducted in Lusaka, Zambia, examining pathogen carriage and enteric enteropathy, which together are hypothesized to contribute to growth faltering in children. A major strength and unique quality of this study is the performance of direct endoscopic measurements of enteric enteropathy in stunted infants. Ethically, it is not possible to perform endoscopy in healthy infants; therefore, a lack of an endoscopy control group is acceptable. The manuscript addresses a critical global health issue. A total of 401 children 0-18 months of age met the anthropometry eligibility requirements and 297 were consented and included in the nutritional rehabilitation study; 213 were still enrolled and followed at 24 months of age. Three groups were compared. First, measurements were made in all 297 children at baseline (median age 11 months). Second, a subset ($n = 119$) of children from these 297, who did not respond to the nutritional intervention, were investigated using esophagogastroduodenoscopy and endoscopy (median age 18 months at assessment). The authors define nutritional intervention non-response as "failure to achieve a positive gradient in LAZ over at least 4 months, together with LAZ consistently below -2 ". Third, a sample of infants ($n = 46$) from the community who exhibited normal growth were recruited as controls from the community (median age 3 months at baseline); these children did not undergo endoscopy.

Children responding to the nutritional intervention did not serve as a comparator. The authors present results for (i) plasma markers of microbial translocation and enteropathy from all non-responders at baseline and at the time of non-response ascertainment and in controls at baseline, (ii) pathogen carriage in all non-responders at baseline and after 3 months and in controls at baseline, (iii) endoscopy in all non-responders at the time of non-response ascertainment, and (iv) RNA sequencing from biopsies (sub-set of non-responders).

The manuscript, especially the endoscopic characterization of stunted children, who did not respond to a nutritional intervention, is a very important contribution to the literature. The hypothesis that the authors suggest is compelling and important. The authors have done a good job responding to the previous reviews. However, a few issues remain.

First, the differences in age distribution between the non-responders and controls can only partially be addressed given the study design. All reviewers mentioned this study design issue. The controls are intended to represent infants without serious growth restriction (> -1 SD of growth curves) and who had not experienced the nutritional intervention. The controls reflect the pathogen carriage and enteropathy and microbial translocation biomarker levels in children 1-5 months of age, while the non-responder case group reflect the pathogen carriage and enteropathy markers of children 1-18 months of age. The controls therefore can only serve as a comparator for the baseline measurements of the youngest non-responders. The authors present a sub-analysis restricted in non-responder cases and controls < 9 months of age, showing that in this sub-set that there were statistically significant differences in LPS, LBP and sCD14 concentrations between the controls and those non-responders measured earliest in life. This strengthens their argument. It is true, as the authors state in their response that "it is important to recognise that the central conclusion of this paper, that microbial translocation, and the host response to it, decline in non-responsive children as they get older despite

sustained pathogen pressure, does not depend on having age-matched controls." However, the structure of the results and some of the analyses obscures the fact that these are only controls for the situation in early life. What if responders (to the nutritional intervention) and/or a set of older controls, for example, had also exhibited decreases in LPS, LBP and sCD14, and increases in iFABP, with a similar pathogen burden over the first 18 months of life? Using the term control group, if the group only serves as a comparison to one fraction of the case group, can complicate the interpretation. The strength of the study lies in the characterization of non-responders to the nutritional intervention, and the control group, given its limitations, makes the inferences a bit more confusing. The authors could consider re-organizing the results so their inferences are clearer to the reader.

Second, if the non-responders were included in an analysis twice (as mentioned in the authors' response, then these 87 children's measurements are not independent. "Response: Yes, the data do contain 87 instances where one child contributed more than one sample. However, it would be incorrect to conclude that age entirely confounds the difference between stunted children and control. We have repeated this analysis stratified by age, and confirm that both age and control status have consistent independent effects on pathogen burden: age increases it and controls have less. Linear regression of total pathogen number shows that controls had on average two fewer pathogens per sample even when adjusted for age; control status and age were highly significant ($P < 0.001$ and $P = 0.005$ respectively)." The issue is a repeated measurement issue, and not entirely an age issue, this needs to be addressed using a repeated measures analysis method (such as GEE). Similarly, the authors state in their revised version, "Multiple linear regression was conducted on square root-transformed LPS, LBP, and sCD14 values with age, breastfeeding and control status as independent variables. These models confirm the independent effect of control status on LPS, LBP and sCD14 when age and breastfeeding were accounted for (Supplementary Table S1)." Does this multiple regression model include baseline measurements for non-responders and controls, or are multiple non-responder measurements included? Again, this will need to be accounted for in a repeated measures analysis. Overall, the statistical analyses in the manuscript might benefit from a review by a statistician.

Third, it would help if the authors moved the text included under the flowchart to the methods. The process for selecting the controls is important.

Reviewer #3 (Remarks to the Author):

I consider that the manuscript does not have any important flaws. The authors have considered the issues raised by the reviewers and responded appropriately and in detail. This has resulted in several modifications, including a more detailed description of the study populations and methods, which have improved the reporting of the findings.

The report builds on the recent focus on enteropathy in stunting. The combination of assessments of biomarkers of epithelial damage, gut leakiness/microbial translocation, nutrient absorption and gut and systemic inflammation as well as intestinal histology and use of the fluorescein dye technique has provided important new insights into the nature of the intestinal lesion, its association with enteropathogens and evolution over time.

The proposal that reduced intestinal surface area, with consequent undernutrition, is an appropriate

adaptive response to sustained pathogen pressure is novel and credible based on the findings of a decrease in biomarkers of mucosal permeability and bacterial translocation but persistence of enteropathogen colonisation, brush border enzymes, nutrient transporter and mucosal barrier function in children with persistent stunting. The study findings and this proposal have direct implications for the prevention and management of malnutrition and, therefore, will be of very great interest to a wide range of people.

To my knowledge, the statistical tests employed are appropriate.

Minor points:

Fig 1: please add median line and error bars in Fig 1.

In Fig 3: It is not clear how to interpret "fraction" on the x axis

Reviewer #4 (Remarks to the Author):

Considerably improved, with reasonable responses to my comments and questions.

Author Rebuttal to Initial comments

Thank you for the additional comments. We have amended the manuscript in response, highlighted in green. Details are given below.

Reviewer 2, point 1: ages of stunted children and controls

The reviewer rightly points out that this is an important issue, identified by all reviewers and clearly stated in the limitations section of our original submission. The reviewer accepts our contention that the overall conclusion of the manuscript does not depend on having age-matched controls, but would like more clarity about the inferences we can draw.

The reviewer goes on to say:

However, the structure of the results and some of the analyses obscures the fact that these are only controls for the situation in early life. What if responders (to the nutritional intervention) and/or a set of older controls, for example, had also exhibited decreases in LPS, LBP and sCD14, and increases in iFABP, with a similar pathogen burden over the first 18 months of life? Using the term control group, if the group only serves as a comparison to one fraction of the case group, can complicate the interpretation. The strength of the study lies in the characterization of non-responders to the nutritional intervention, and the control group, given its limitations, makes the inferences a bit more confusing. The authors could consider re-organizing the results so their inferences are clearer to the reader.

We fully agree that the control group is only a control group for the first 9 months of life, up to which age we have a case-control comparison as in this age group as only baseline samples were collected. We have added this sentence to the Summary:

Control children (aged 1-9 months) contributed data only at baseline.

and in the Results we have made it clear that the comparisons between cases and controls include only baseline samples collected below 9 months of age (page 4). This has been added to the Methods (page 15). Figures 2 and 3 also help make this clear.

Reviewer 2, point 2: repeated measures analysis

The regression analysis was performed at the suggestion of the reviewer, and the reviewer has correctly observed that it would be inappropriate to include repeated measures in a regression without using a statistical technique which allows for this, such as GEE. Dr VanBuskirk, penultimate author and our study statistical adviser, has reviewed the reviewer's comments and agrees with them. However, it is crucial to emphasise that the controls were only sampled once (at 1-9 months of age), and the cases twice (at baseline and at non response). GEE will therefore not overcome the limitation of having controls only for young children. Instead, she recommended conducting Kruskal-Wallis testing of pathogen burdens in cases and controls, restricted to children under 9 months of age (therefore only baseline samples), and linear regression of total pathogen number and logistic regression of individual pathogens (Supplementary Table S2). Linear regression of square-root transformed biomarkers was also performed, again restricted to children under 9 months of age. These analyses again confirm that controls have significantly fewer pathogens than cases in the same age range, and significantly higher circulating concentrations of LPS, LBP and sCD14. We have re-worded this in the Results (page 5) to make this clearer, and have included appropriate description in Methods.

Reviewer 2, point 3

We have moved the detail from legend to Methods (page 12).

Reviewer 3, point 1

We have changed this figure to a box plot to make the median and quartiles easier to see.

Reviewer 3, point 2

We have changed fraction to proportion on the x axis and explained it fully in the legend.

Decision Letter, first revision:

Dear Professor Kelly,

Thank you for your patience while your manuscript "Adaptation of the small intestine to intestinal pathogens in Zambian children with stunting" was under peer review at Nature Microbiology. It has now been seen by our referee, and in the light of reviewer advice I am delighted to say that we can in principle offer to publish it.

{REDACTED}}

We would like you to revise your paper to address the points made by the reviewers, and to ensure that it is in Nature Microbiology format. Please use the lightly edited version (attached) with my queries and guidance in (using track changes) to form the basis of the final main text document.

There are no remaining reviewer comments . Editorially, we will need you to make some changes so that the paper complies with our Guide to Authors at <http://www.nature.com/nmicrobiol/info/gta>.

Specific points:

In particular, while checking through the manuscript and associated files, we noticed the following specific points which we will need you to address:

1. Thanks for providing a dryad link for the data in this study. Also note that per our policies all sequencing reads should be in the SRA or similar with an accession number.
2. Please insert short titles for the figures.
3. Extended data. Per journal guidelines, we use "Extended Data". Please see below for additional information on how to format and refer to Extended Data. We can allow a maximum of 10 extended data figures and so we would suggest that you convert the 6 Supplementary figures into Extended Data Figures and rename throughout the manuscript. ED figures (see below) will show in the HTML.
4. The main and extended data figures need to be uploaded individually and without captions; they should be provided in eps, tiff or jpeg format. The main text file needs to be submitted in Word or LaTeX format, and all (main and extended) figure legends need to be included at the end of this file. Do NOT include any figures in the main text file.

General points:

We will also need you to check through all of the following general points when preparing the final version of your manuscript:

5. Titles should give an idea of the main finding of the paper and ideally not exceed 90 characters (including spaces). We discourage the use of active verbs and do not allow punctuation. See my suggestion

6. Choosing the right electronic format for your figures at this stage will speed up the processing of your paper. We would like the figures to be supplied as vector files - EPS, PDF, AI or postscript (PS) file formats (not raster or bitmap files), preferably generated with vector-graphics software (Adobe Illustrator for example). Please try to ensure that all figures are non-flattened and fully editable. All images should be at least 300 dpi resolution (when figures are scaled to approximately the size that they are to be printed at) and in RGB colour format. Please do not submit Jpeg or flattened TIFF files. Please see also 'Guidelines for Electronic Submission of Figures' at the end of this letter for further detail.

Please view http://www.nature.com/authors/editorial_policies/image.html for more detailed guidelines.

7. We will edit your figures/tables electronically so they conform to Nature Microbiology style. If necessary, we will re-size figures to fit single or double column width. If your figures contain several parts, the parts should be labelled lower case a, b, and so on, and form a neat rectangle when assembled.

8. Please check the PDF of the whole paper and figures (on our manuscript tracking system) VERY CAREFULLY when you submit the revised manuscript. This will be used as the 'reference copy' to make sure no details (such as Greek letters or symbols) have gone missing during file-transfer/conversion and re-drawing.

9. All Supplementary Information must be submitted in accordance with the instructions in the attached Inventory of Supporting Information, and should fit into one of three categories:

1. **EXTENDED DATA:** Extended Data are an integral part of the paper and only data that directly contribute to the main message should be presented. These figures will be integrated into the full-text HTML version of your paper and will be appended to the online PDF. There is a limit of 10 Extended Data figures, and each must be referred to in the main text. Each Extended Data figure should be of the same quality as the main figures, and should be supplied at a size that will allow both the figure and legend to be presented on a single legal-sized page. Each figure should be submitted as an individual .jpg, .tif or .eps file with a maximum size of 10 MB each. All Extended Data figure legends must be provided in the attached Inventory of Accessory Information, not in the figure files themselves.

2. **SUPPLEMENTARY INFORMATION:** Supplementary Information is material that is essential background to the study but which is not practical to include in the printed version of the paper (for example, video files, large data sets and calculations). Each item must be referred to in the main manuscript and detailed in the attached Inventory of Accessory Information. Tables containing large data sets should be in Excel format, with the table number and title included within the body of the

table. All textual information and any additional Supplementary Figures (which should be presented with the legends directly below each figure) should be provided as a single, combined PDF. Please note that we cannot accept resupplies of Supplementary Information after the paper has been formally accepted unless there has been a critical scientific error.

All Extended Data must be called out in your manuscript and cited as Extended Data 1, Extended Data 2, etc. Additional Supplementary Figures (if permitted) and other items are not required to be called out in your manuscript text, but should be numerically numbered, starting at one, as Supplementary Figure 1, not SI1, etc.

3. SOURCE DATA: We encourage you to provide source data for your figures whenever possible. Full-length, unprocessed gels and blots must be provided as source data for any relevant figures, and should be provided as individual PDF files for each figure containing all supporting blots and/or gels with the linked figure noted directly in the file. Statistics source data should be provided in Excel format, one file for each relevant figure, with the linked figure noted directly in the file. For imaging source data, we encourage deposition to a relevant repository, such as figshare (<https://figshare.com/>) or the Image Data Resource (<https://idr.openmicroscopy.org>).

10. Nature Research journals [encourage authors to share their step-by-step experimental protocols](https://www.nature.com/nature-research/editorial-policies/reporting-standards#protocols) on a protocol sharing platform of their choice. Nature Research's Protocol Exchange is a free-to-use and open resource for protocols; protocols deposited in Protocol Exchange are citable and can be linked from the published article. More details can found at www.nature.com/protocolexchange/about.

11. Please note that after the paper has been formally accepted you can only provide amended Supplementary Information files for critical changes to the scientific content, not for style. You should clearly explain what changes have been made if you do resupply any such files.

12. Figure legends must provide a brief description of the figure and the symbols used, within 350 words. This must include definitions of any error bars employed in the figures.

13. It is a condition of publication that you include a statement before the acknowledgements naming the author to whom correspondence and requests for materials should be addressed.

14. Please use the attached checks on statistics and presentation, made by our partner SNTPS, to ensure stats are correctly presented and reported. Please also use these comments to finalise a reporting summary, which should be included in the final submission and which will be published with your paper.

15. Finally, we require authors to include a statement of their individual contributions to the paper -- such as experimental work, project planning, data analysis, etc. -- immediately after the acknowledgements. The statement should be short, and refer to authors by their initials. For details please see the Authorship section of our joint Editorial policies at http://www.nature.com/authors/editorial_policies/authorship.html

If the revised paper is in Nature Microbiology format, in accessible style and of appropriate length, we

shall accept it for publication immediately.

Please resubmit electronically

- * the final version of the text (not including the figures) in either Word or Latex.
- * publication-quality figures. For more details, please refer to our Figure Guidelines, which is available here: https://mts-nmicrobiol.nature.com/letters/Figure_guidelines.pdf
- * Extended Data & Supplementary Information, as instructed
- * a point-by-point response to any issues raised by our referees and to any editorial suggestions.
- * any suggestions for cover illustrations, which should be provided at high resolution as electronic files. Please note that such pictures should be selected more for their aesthetic appeal than for their scientific content. I am sure you will understand that we cannot make any promise as to whether any of your suggestions might be selected for the cover of Nature Microbiology.

Please use the following link to access your home page:

{REDACTED}

* This url links to your confidential homepage and associated information about manuscripts you may have submitted or be reviewing for us. If you wish to forward this e-mail to co-authors, please delete this link to your homepage first.

Please also send the following forms as a PDF by email to microbiology@nature.com.

* Please sign and return the <http://www.nature.com/documents/snl-ltp.docx> target="_blank">Licence to Publish form .

* Or, if the corresponding author is either a Crown government employee (including Great Britain and Northern Ireland, Canada and Australia), or a US Government employee, please sign and return the <http://www.nature.com/documents/snl-ltp-crown.docx> target="_blank"> Licence to Publish form for Crown government employees, or a <http://www.nature.com/documents/snl-ltp-govus.docx> target="_blank"> Licence to Publish form for US government employees.

* Should your Article contain any items (figures, tables, images, videos or text boxes) that are the same as (or are adaptations of) items that have previously been published elsewhere and/or are owned by a third party, please note that it is your responsibility to obtain the right to use such items and to give proper attribution to the copyright holder. This includes pictures taken by professional photographers and images downloaded from the internet. If you do not hold the copyright for any such item (in whole or part) that is included in your paper, please complete and return this <http://www.nature.com/documents/thirdpartyrights-origres.doc> target="_blank">Third Party Rights Table, and attach any grant of rights that you have collected.

For more information on our licence policy, please consult <http://npg.nature.com/authors>.

ORCID

Nature Microbiology is committed to improving transparency in authorship. As part of our efforts in this direction, we are now requesting that all authors identified as 'corresponding author' create and link their Open Researcher and Contributor Identifier (ORCID) with their account on the Manuscript Tracking System (MTS) prior to acceptance. ORCID helps the scientific community achieve unambiguous attribution of all scholarly contributions. For more information please visit <http://www.springernature.com/orcid>

For all corresponding authors listed on the manuscript, please follow the instructions in the link below to link your ORCID to your account on our MTS before submitting the final version of the manuscript. If you do not yet have an ORCID you will be able to create one in minutes. <https://www.springernature.com/gp/researchers/orcid/orcid-for-nature-research>

IMPORTANT: All authors identified as 'corresponding author' on the manuscript must follow these instructions. Non-corresponding authors do not have to link their ORCIDs but are encouraged to do so. Please note that it will not be possible to add/modify ORCIDs at proof. Thus, if they wish to have their ORCID added to the paper they must also follow the above procedure prior to acceptance.

To support ORCID's aims, we only allow a single ORCID identifier to be attached to one account. If you have any issues attaching an ORCID identifier to your MTS account, please contact the Platform Support Helpdesk.

Nature Research journals encourage authors to share their step-by-step experimental protocols on a protocol sharing platform of their choice. Nature Research's Protocol Exchange is a free-to-use and open resource for protocols; protocols deposited in Protocol Exchange are citable and can be linked from the published article. More details can found at www.nature.com/protocolexchange/about.

We hope that you will support this initiative and supply the required information. Should you have any query or comments, please do not hesitate to contact me.

{REDACTED}

Reviewer Expertise:

Reviewer #2 (Remarks to the Author):

There are no follow-up remarks for the authors.

Final Decision Letter:

Dear Professor Kelly,

I am pleased to accept your Article "Adaptation of the small intestine to microbial enteropathogens in Zambian children with stunting" for publication in Nature Microbiology. Thank you for having chosen to submit your work to us and many congratulations.

Before your manuscript is typeset, we will edit the text and look particularly carefully at the titles of all papers to ensure that they are relatively brief and understandable.

The subeditor may send you the edited text for your approval. Once your manuscript is typeset you will receive a link to your electronic proof via email within 20 working days, with a request to make any corrections within 48 hours. If you have queries at any point during the production process then please contact the production team at rjsproduction@springernature.com. Once your paper has been scheduled for online publication, the Nature press office will be in touch to confirm the details.

Acceptance of your manuscript is conditional on all authors' agreement with our publication policies (see www.nature.com/nmicrobiolate/authors/gta/content-type/index.html). In particular your manuscript must not be published elsewhere and there must be no announcement of the work to any media outlet until the publication date (the day on which it is uploaded onto our website).

Your paper will be OPEN ACCESS.

We welcome the submission of potential cover material (including a short caption of around 40 words) related to your manuscript; suggestions should be sent to Nature Microbiology as electronic files (the image should be 300 dpi at 210 x 297 mm in either TIFF or JPEG format). Please note that such pictures should be selected more for their aesthetic appeal than for their scientific content, and that colour images work better than black and white or grayscale images. Please do not try to design a

cover with the Nature Microbiology logo etc., and please do not submit composites of images related to your work. I am sure you will understand that we cannot make any promise as to whether any of your suggestions might be selected for the cover of the journal.
